# Ilama VHH as a Substitute for Rabbit Polyclonal Antibodies in ELISpot Application

**DOI:** 10.3390/ijms262411881

**Published:** 2025-12-09

**Authors:** Chloé Reynas, Jérémy Balland, Harmonie Simonin, Pierre-Emmanuel Baurand

**Affiliations:** Diaclone SAS—Part of Medix Biochemica Group, 6 Rue Dr Jean-François-Xavier Girod, 25000 Besançon, France; chloe.reynas@medixbiochemica.com (C.R.); jeremy.balland@medixbiochemica.com (J.B.); harmonie.simonin@medixbiochemica.com (H.S.)

**Keywords:** Ilama, VHH, phage display, ELISpot, ELISA, substitute, polyclonal, mIFN-γ

## Abstract

Enzyme-Linked-Immunosorbent-Spot (ELISpot) is a highly sensitive technique capable of detecting low-level immune responses, offering critical insights into therapy-induced immune activation. Our mouse interferon-gamma (mIFN-γ) ELISpot assay was originally based on a monoclonal capture antibody and a rabbit polyclonal detection antibody. The objective of our study was to replace the polyclonal detection antibody with a monoclonal alternative, using a llama immune library and phage display technology. A llama was immunized with recombinant mIFN-γ, and an immune VHH library was constructed. The library underwent two rounds of panning using the recombinant antigen. Subsequently, 190 clones were screened by Enzyme-Linked-Immunosorbent Assay (ELISA), yielding 27 specific binders to mIFN-γ. Sequence analysis revealed 24 unique clones grouped into four families based on their CDR3-VH sequences. One representative clone from each family was reformatted as VHH-Human Fragment Crystallizable (VHH-hFc) fusion and produced recombinantly for testing in the ELISpot assay. The purified candidates were evaluated in pairs on native mIFN-γ from mouse splenocytes. Two candidates, H3 and G4, were selected for further trial. Comparative analysis of ELISpot performance showed that G4 is a promising substitute for the original rabbit polyclonal antibody, enhancing the overall performance of the mIFN-γ ELISpot assay. This study highlights the potential of VHH antibodies in ELISpot applications and supports their use as a robust, reproducible alternative to polyclonal antibodies.

## 1. Introduction

Cytokines are essential proteins that mediate communication between immune cells, playing a central role in coordinating and regulating immune responses. Among them, interferon gamma (IFN-γ) is a powerful conductor of the immune system, orchestrating the body’s defense against invaders and shaping immune responses [1].

IFN-γ is mainly produced by Natural Killer (NK) cells and T lymphocytes, particularly Th1 CD4^+^ and CD8^+^ cytotoxic T cells, in response to antigenic stimulation. In the innate immune response, it enhances the cytotoxic function of NK cells and contributes to macrophage activation [2]. IFN-γ also promotes the differentiation of naive CD4^+^ T cells into Th1 cells, which are essential for cell-mediated immune responses. Additionally, it up-regulates the expression of major histocompatibility complex molecules on antigen-presenting cells, facilitating efficient antigen presentation and subsequent T cell activation [3].

Its involvement in immune responses makes it a central player in various pathological conditions, including infectious diseases, autoimmune disorders, and cancer [4]. However, while IFN-γ is vital for host defense, its dysregulation can contribute to chronic inflammation and immune-related diseases. Therefore, understanding and accurately assessing IFN-γ secretion by immune cells is of major importance in both research and clinical diagnostics.

A common method to evaluate IFN-γ secretion by immune cells in vitro is the Enzyme-Linked-Immunosorbent-Spot (ELISpot) assay. Originally developed to detect antibody-secreting cells through a combination of Enzyme-Linked-Immunosorbent Assay (ELISA) and immunoblotting principles, the ELISpot assay has since been optimized to measure cytokine secretion at the single-cell level. This highly sensitive technique can detect low-level immune responses and provides valuable insights into therapy-induced immune activation. Today, ELISpot assays are widely used to evaluate the efficiency of vaccines or immunotherapies, with a large number of ELISpot kits available in the market [5].

However, while ELISpot is a very sensitive assay for monitoring immune responses, its accuracy and reproducibility critically depend on the quality of antibodies used. In this context, phage display provides an ideal platform for generating high-affinity antibodies. This recombinant technique relies on displaying large libraries of antibody fragments on the surface of bacteriophages, allowing for the selection of binders with strong specificity and affinity against a target of interest [6]. By iteratively enriching phages that bind the antigen, phage display enables the rapid isolation of highly specific and affine monoclonal antibodies.

Different antibody fragments can be displayed on the phage surface, including Fab, scFv, and VHHs. VHHs are single-domain antibody fragments (size~15 kDa), also called nanobodies, discovered in camelids (camels, llamas, and alpacas). Unlike conventional antibodies, VHHs consist of only heavy chains, lacking light chains. As conventional monoclonal antibodies (mAbs), VHH heavy chains are composed of four framework regions (FR1 to FR4) spaced by three complementarity-determining regions (CDR1, CDR2, CDR3), with CDR3 being longer than in standard mAbs and playing a major role in antigen recognition [7]. In addition to their small size (1/10th of a full antibody), VHHs present advantages such as high stability (pH, temperature), good solubility, and low immunogenicity, and they are easily engineerable for humanization. These properties make them valuable for therapeutic use, diagnostic tools, and research applications [8] in various fields such as infectious diseases [9], imagery (as an in vivo imaging agent), or cancer therapy. To our knowledge, VHHs have not been used as a raw material in ELISpot applications.

At Diaclone, part of the Medix Biochemica group, our mIFN-γ ELISpot kit is based on a monoclonal capture antibody and a rabbit polyclonal detection antibody. However, the polyclonal reagent is subject to interbatch variability and potential supply limitations. The objective of this project is to develop a monoclonal antibody substitute for the current rabbit polyclonal detection antibody using a llama immune library combined with phage display technology. This new monoclonal detection antibody will ensure a more sustainable, reliable, and controlled supply for our mouse IFN-γ ELISpot kit.

## 2. Results

### 2.1. Immunization

To generate VHHs specific to mIFN-γ, llamas were immunized with our in-house produced mIFN-γ histag recombinant antigen (ref. 210127, Diaclone, Besançon, France). To assess the immune response of llamas, serum samples were collected before immunization (Day 0, D0) and at the end of the immunization protocol (Day 40, D40). The evaluation was performed using ELISA, using both our in-house produced recombinant mIFN-γ antigen (ref. 210127, Diaclone, Besançon, France) and a commercial recombinant mIFN-γ protein (ref. 315-05, PeproTech, Rocky Hill, NJ, USA). For each time point, sera were diluted in six 10-fold serial dilutions ranging from 1:100 to 1:10,000,000. At D40, a strong antigen-specific response was observed, with high optical density values (OD_450nm_) detected across a wide range of serum dilutions (from 1:1000 to 1:100,000), indicating the successful generation of high-titer antibodies. In contrast, pre-immune sera (D0) showed minimal reactivity to either form of mIFN-γ (Figure 1).

### 2.2. Library Construction

A VHH phage display library was constructed using peripheral blood mononuclear cells collected from the llama immunized with our in-house recombinant mouse IFN-γ antigen (ref. 210127, Diaclone, Besançon, France). Total ribonucleic acid (RNA) was extracted from Peripheral Blood Mononuclear Cells (PBMCs), and complementary deoxyribonucleic acid (cDNA) was synthesized by reverse transcription. VHH-coding sequences were amplified by Polymerase Chain Reaction (PCR) using cDNA as the template. The VHH phage display library was constructed by cloning the VHH gene repertoire into the pCOMB3 phagemid vector.

Library size was estimated by titration and reached 2 × 10^6^ colony-forming units per mL (CFU/mL). Correct insert presence and size in the phagemid were evaluated by colony PCR on 48 randomly selected clones, 100% of which contained an insert of the expected size (~800 bp), indicating a high insertion efficiency. To assess the quality of insert cloning in phagemid, the presence of open reading frame (ORF) was evaluated by analyzing the exact sequences of 10 individual clones by Sanger sequencing. All sequences had the expected VHH variable domain structure, characterized by an alternating arrangement of four conserved FRs that ensure structural stability and three hypervariable CDRs forming the antigen-binding sites. All 10 clones were unique, with distinct CDR3 sequences. Collectively, these results confirm the successful construction of a high-quality VHH library, characterized by a large size (>10^6^ CFU/mL), high insertion rate (>95%), and sequence diversity (Table 1), thereby providing a reliable starting point for the selection of mouse IFN-γ-specific VHH.

### 2.3. Panning—Selection

To isolate and amplify phages expressing VHHs capable of specific and strong binding to mIFN-γ, two rounds of panning were performed on our immobilized in-house mIFN-γ (ref. 210127, Diaclone, Besançon, France) from the initial immune VHH phage display library. To enhance both binding specificity and affinity, the second panning round was conducted under more stringent conditions: the phages-mIFN-γ incubation time was reduced from 1 h in the first panning round to 30 min in the second, and the number of wash steps was increased from 8 to 15. Panning efficiency was assessed by titration of the eluted mIFN-γ binding phages after each round.

To evaluate the efficiency of the panning process, we calculated a specific enrichment for each round, defined as the ratio between the (output/input) of phage eluted after exposure to mIFN-γ and the corresponding (output/input) obtained after incubation with Phosphate-Buffered Saline-Bovine Serum Albumine 1% (PBS-BSA 1%) corresponding to background binding. Dividing output by input reflects the proportion of phage that were able to bind and be recovered under each condition. By comparing this proportion between the mIFN-γ condition and the PBS–BSA 1% background condition, the enrichment factor indicates how effectively the selection process increases the representation of phage specifically binding to mIFN-γ.

In Round 1, the mIFN-γ exposition yielded 1.5 × 10^5^ phages from an input of 1.4 × 10^11^, whereas the PBS–BSA 1% control produced 3 × 10^2^ phages, corresponding to a specific enrichment of 500. This indicates an initial selection of mIFN-γ–specific clones. In Round 2, despite a lower input (2.3 × 10^9^ phages), the mIFN-γ exposition produced 1.2 × 10^5^ phages, while the PBS–BSA 1% yielded a single colony. The resulting enrichment in Round 2 increased to 1.2 × 10^5^, representing a 240-fold improvement compared with Round 1 (Table 2). These results demonstrate that the biopanning process efficiently selected and amplified phages specifically binding mIFN-γ, while non-specific binders were effectively eliminated following only two rounds of selection.

### 2.4. VHH Screening by ELISA

After two rounds of panning, 95 individual VHH clones from round 1 and 95 from round 2 were expressed in *E. coli* TG1 cells following isopropyl β-D-1-thiogalactopyranoside (IPTG) induction. The VHHs were produced in the periplasmic space, and their expression was confirmed by ELISA on periplasmic extracts. These periplasmic extracts containing VHHs were subsequently used for screening to evaluate direct binding to our own recombinant murine IFN-γ (ref. 210127, Diaclone, Besançon, France) and to assess their ability to be used as detection mAbs in a sandwich ELISA set-up, using an anti-rat IFN-γ commercial antibody cross-reacting with murine IFN-γ in vitro and in vivo (Clone DB-1, ref. CT032, U-Cytech, Utrecht, The Netherlands) as the capture antibody. The DB-1 clone is the capture antibody employed in our mIFN-γ ELISpot kit (ref. 862.031.xxx, Diaclone, Besançon, France).

Out of the 190 screened clones, 14% (27 clones) were identified as specific binders to murine IFN-γ (Figure 2). The remaining clones (86%) cross-reacted on irrelevant histag antigen. These clones exhibited an ELISA signal (OD_450nm_) at least three times higher than the background (PBS-BSA 1%), either in direct ELISA against immobilized mIFN-γ and/or in sandwich ELISA using the DB-1 capture antibody mIFN-γ complex. These 27 binders include 13 and 14 binders from panning rounds 1 and 2 (R1 and R2). Out of the 27 specific VHHs, there were three distinct binding profiles: 52% of the clones (14/27) were able to recognize murine IFN-γ both when directly immobilized on the plate and when in complex with the DB-1 antibody. A total of 33% of the clones (9/27) bound murine IFN-γ only when presented by DB-1 antibody, and 15% of the clones (4/27) were positive only to immobilized mIFN-γ, which may reflect recognition of epitopes that are masked or conformationally altered in the sandwich set-up (Figure 2). These results highlight the efficient selection of high-affinity VHHs against mIFN-γ after only one or two panning rounds, while also demonstrating the functional diversity of the isolated VHHs in terms of epitope recognition and affinity.

### 2.5. Identification of VHH Clones

Following two rounds of panning, the 27 best-performing VHH candidates recognizing murine IFN-γ alone and/or in complex with DB-1 were selected for DNA sequencing to determine their nucleotide and amino acid sequences and to eliminate redundant clones. Among them, 26 encoded complete VHHs, while 1 clone (R1-E10) was non-coding and excluded from further analysis. Redundant clones were identified based on sequence identity: R2-G10 and R2-D2, as well as R1-C1 and R1-H1, were found to be identical; only the first clone of each pair was retained. In total, 24 unique coding VHHs were identified (differing by one amino acid). Among these, twelve were considered unique due to somatic hypermutations within the framework regions. The 24 VHHs were grouped into families based on their CDR3 sequences, as this region is typically described as the most critical for antigen binding. Four distinct CDR3 families were defined; the CDR3 sequences are shown in Table 3. The top-performing clone from each family, as determined during the screening, was selected as the representative and produced in recombinant form for further characterization.

### 2.6. Reformatting and Production of Recombinant VHH-hFC

Following engineering by molecular biology techniques to reformat VHH into a recombinant VHH-hFC fusion antibody, pilot batches were produced during 14 days in 10 mL of transfected Chinese hamster ovary (CHO) cells. Then, supernatants were purified using protein A affinity chromatography. As shown in Table 4, yields from transient CHO cell transfections ranged from 0.5 to 1.25 mg of purified VHH-hFc, which was then used for the initial pairing tests and evaluation in an ELISpot assay.

After a first evaluation, only VHH-G4 and VHH-H3 were retained (see Section 2.7). To continue evaluating these two candidates, new larger-scale 100 mL productions were performed. A total of 14 for VHH-G4 and 16 mg for H3 were obtained after purification. The quality of the purified VHH-hFC was visually assessed by Sodium Dodecyl Sulfate–PolyAcrylamide Gel Electrophoresis (SDS-PAGE) (4–15%), confirming a high purity as shown in Figure 3.

The theoretical molecular weight range for VHH-hFC is 40 kDa. The dimeric expected sizes are theoretically near 80 kDa. Under reducing conditions, the two candidates show fragments around 40 kDa, consistent with monomeric VHH-hFC. Under non-reducing conditions, predominantly dimeric forms are observed, migrating at approximately 80–90 kDa (Figure 3).

### 2.7. Antibody Pairing Tests

#### 2.7.1. ELISA Pairing Tests

After purification, VHH-hFC antibodies were biotinylated and evaluated in ELISA as capture or detection (biotinylated) antibodies, with native mIFN-γ antigen (from splenocyte culture supernatant). We observed that VHH-D12 and VHH-G12 give no specific signal when using DB-1 as a capture or detection antibody. Therefore, VHH-D12 and VHH-G12 were excluded from further testing.

As shown in Table 5, on native mIFN-γ, effective antibody pairings were obtained using H3-biotin and G4-biotin as detection antibodies with DB-1 as the capture antibody. Similarly, effective combinations were observed when H3 or G4 were used as capture antibodies with DB-1-biotin for detection. Based on this classical pairing assay, the performance of H3 and G4 as detection antibodies appears comparable to that of the polyclonal antibody. However, the G4-H3 combination does not form a functional pair, regardless of the assembly of antibodies (capture or detection).

#### 2.7.2. ELISpot Pairing Test

After an ELISA pairing validation, another pairing test was performed in the ELISpot application using mouse splenocytes (Figure 4) to assess whether the VHH-hFC candidates could serve as an alternative to the currently used rabbit polyclonal detection antibody. Cells were incubated on an ELISpot plate overnight with (Stimulated, S) or without (Unstimulated, US) the immune cell stimulant Phorbol Myristate Acetate (PMA) and Ionomycin. The results show that H3-Biot and G4-Biot exhibited a strong signal when paired with DB-1 used as the capture antibody. The magnitude of the response was too high to allow accurate spot enumeration, but a pronounced purple coloration was clearly visible on the membrane in wells containing the stimulated cells compared to wells containing unstimulated cells. In addition, H3 as the capture antibody shows high background noise as evidenced by strong membrane coloration in the unstimulated wells, regardless of the detection antibody used (DB-1 or G4-Biot). Thus, H3 and G4 can be paired together with G4 as the capture and H3 as the detection antibody (biotinylated), but the signal remains lower than our reference (DB-1/Polyclonal-Biot).

Based on those results, it was decided to keep DB-1 as the capture antibody and continue to test H3 and G4 as detection antibodies at different concentrations. With optimal concentration (25 ng/well), both antibodies showed similar results with a better sensitivity than the rabbit polyclonal antibody, with mean values of 301 and 330 spots, respectively, compared to 147 spots for the polyclonal (Figure 5). Both options (DB-1/H3_Biot or DB-1/G4_Biot) show a better performance than our previous pair (DB-1/Polyclonal_Biot).

### 2.8. New Production and Stability Assessment of VHH-hFC

To determine which of the two VHH candidates should be retained as the substitute, a new transient transfection production in CHO cells was performed (see Section 2.6). These new batches enabled interbatch comparisons and stability tests.

Newly produced antibody batches were subjected to varied storage conditions: 4 °C, room temperature (RT), 37 °C, or −80 °C. After one week of incubation, antibodies were tested again as detection mAbs in our ELISpot kit. Then, the coefficient of variation (CV) was calculated by dividing the standard deviation by the mean value of spots multiplied by 100 (expressed as a percentage). For the same sample, a lower CV indicates better reproducibility and stability. The reference sample used is the first batch of VHH-hFC stored at 4 °C (in PBS-BSA 1%–Azide 0.09%).

As shown in Table 6, the batch-to-batch reproducibility is first assessed by a comparison between the two batches (reference and newly produced) stored at 4 °C. In this case, H3 shows a higher CV (19%) than G4 (5%), suggesting that H3 gives more variable results compared to G4. Then, the storage conditions applied seem to affect H3 stability, with its CV increasing from 18% at RT to 24% at 37 °C and 27% at −80 °C. The G4 antibody stays stable with CV between 9% and 8% at RT, 37 °C or −80 °C. Based on this stability test, it appears that G4 is more stable (8% mean CV versus 22% for H3) and provides more reproducible results. Based on these findings, the G4 candidate was selected as the final substitute for polyclonal use in our ELISpot.

Finally, we compared the sensitivity and linearity of our ELISpot assay using the previous antibody pair (DB-1/Polyclonal_Biot) versus the new pair (DB-1/G4_Biot). Sensitivity was assessed by the number of spots detected, while linearity was evaluated by the correlation between the number of stimulated cells and the detected spots. As shown in Figure 6, the new antibody pair demonstrated improved sensitivity, detecting approximately 650 spots compared to 350 spots for the previous pair using 100,000 stimulated cells, and exhibited better linearity, with an R^2^ of 0.99 versus 0.96 for the previous pair.

## 3. Discussion

Polyclonal antibodies (pAbs) are a heterogeneous mixture of antibodies produced in response to an antigen. In most cases, these antibody pools recognize multiple epitopes on the same antigen, providing them with high sensitivity and making them particularly effective for detecting weakly immunogenic antigens, as they amplify the immune response compared to mAbs [10]. Another advantage of pAbs lies in their rapid and cost-effective production.

However, their production depends on the lifespan of the immunized animal, which limits long-term supply. In addition, batch-to-batch variability is a major drawback, as each immunization yields a different antibody composition, reducing reproducibility and potentially affecting performance [10].

The present work describes the development of VHH candidates to assess their potential as substitutes for polyclonal antibodies in specific applications such as ELISpot kits. To the best of our knowledge, this is the first report demonstrating the replacement of polyclonal antibodies with one VHH in an ELISpot assay.

The development of mAbs from the llama immune library as shown in this study was highly successful.

Firstly, the overall quality of the generated immune VHH library is excellent, with a very high insertion rate (>95%) and a size of 2 × 10^6^ CFU/mL. These parameters are comparable to those reported in previous studies using mouse ScFv libraries [11] or rabbit/human chimeric Fab libraries [12].

Secondly, the panning process using phage-displayed VHHs was also highly effective. Two rounds of selection on the target antigen were sufficient to achieve strong enrichment of specific phages from the immune library. These promising results are consistent with those obtained using recombinant antigens with VHH [13], chimeric Fab libraries [12], or even with cell-based biopanning using ScFv libraries [11]. Notably, this is the first time for us such a high level of enrichment has been achieved after just a single round of panning.

Thirdly, the screening process led to the identification of 27 specific binders, representing 14% of all tested candidates. These included 13 and 14 binders from panning rounds 1 and 2 (R1 and R2), respectively. Sequencing revealed 24 unique binders, grouped into four distinct families. This represents one-eighth of the total screened candidates, with an even distribution between R1 and R2. To our experience, this is the first time such a high yield of unique binders has been obtained from the first round of panning. Overall, the panning and screening results clearly demonstrate that VHH-based projects are easier to conduct and more efficient than Fab or ScFv-based approaches.

Fourthly, the recombinant reformatting of final candidates H3 and G4 was performed using a VHH-hFc design, aiming to enhance the overall stability of the VHHs by improving their half-life, as previously reported by Zhang et al. [14].

Production yields obtained after purification (>100 µg/mL) were conforms to yields already reported in mammalian systems [8] and were slightly higher than those typically achieved for standard monoclonal antibodies produced transiently in CHO systems.

Finally, in ELISpot applications, the superior performance of VHH-hFc over the polyclonal antibody is clearly demonstrated. Furthermore, recombinant G4 candidates provide a sustainable solution to overcome the supply and performance issues commonly encountered with pAb-based detection antibodies.

## 4. Materials and Methods

### 4.1. Immunization

Llama immunization was outsourced (Llama farm, Mamirolle, France). One female llama was immunized in a 42-day protocol with our own recombinant murine IFN-γ antigen produced in CHO cells, (Accession number NP_032363.1, His23-Cys155). Six weekly subcutaneous injections of antigen mixed with incomplete Freund’s adjuvant (ref. Sigma F5506, Merck KGaA, Darmstadt, Germany) were performed on day 0, 7, 14, 21, 28 and 35. The first two injections were performed with 200 µg of antigen, and the rest were performed with 100 µg. Five days after the last injection (day 42), 250 mL of blood was collected from immunized animals and used for subsequent molecular biology processes (RNA isolation, cDNA synthesis, and library construction).

### 4.2. Library Construction

Total RNA isolation was performed using blood cells (3.5 × 10^8^) from immunized llamas with the Rneasy Maxi kit (ref. 75162, Qiagen, Hilden, Germany) following the manufacturer’s instructions. RNA was converted into cDNA with the SuperScript III Reverse Transcriptase (ref. 18080051, Invitrogen, Waltham, MA, USA) based on the manufacturer’s instructions. The RNA quality was assessed via the Experion™ Automated Electrophoresis System (Biorad Laboratories, Hercules, CA, USA). Highly pure RNA was obtained with an RNA quality indicator of 9.9/10.

VHH sequences were amplified following two rounds of successive PCRs using the same couples of primers adapted from Pardon Et Al. [15]. The first PCR rounds were performed to amplify VHs and VHH sequences starting from cDNA. Amplifications were performed with Phusion DNA Polymerase (ref. F530L, Thermo Fisher Scientific, Waltham, MA, USA) using the following PCR program: 94 °C for 2 min; 30 cycles of 94 °C for 30 sec, 60 °C for 30 sec, 72 °C for 60 s, and final amplification at 72 °C for 7 min. The second PCR rounds were performed on gel purified 700 bp PCR products (corresponding to heavy chain antibody repertoire), following the PCR program: 94 °C for 2 min; 20 cycles of 94 °C for 30 s, 62 °C for 30 s, 72 °C for 60 sec; and final amplification at 72 °C for 7 min. This second PCR round only serves to amplify the VHH variable domain and to add 20 bp recombination parts on the extremities of VHH fragments amplified from the first PCR round. A VHH sequence library was built by recombination cloning (NEBuilder^®^, ref. E2621L, New England Biolabs, Ipswich, MA, USA). Purified VHH PCR products (~500 bp) were cloned in the pCOMB3 vector between *Pst*I and *Not*I restriction sites, and a C-terminal His tag, followed by a Myc tag, was added to facilitate protein purification and detection (Figure 1). The library was finally constructed by electroporation of the cloned phagemids into electrocompetent TG1 cells (ref. 6052-2, Lucigen, Middeleton, WI, USA).

### 4.3. Library Validation

Validation was performed using the same process with some adjustments described by Baurand et al. [11,12]. Briefly, 5 µL spots of 10–10^6^ diluted library were applied on Lennox broth (LB) agar plates and incubated overnight at 37 °C. On the following day, the number of colonies forming within each spot was counted to determine the library size. The PCR-product insertion (in %) in the phagemid was evaluated by colony PCR on 48 randomly chosen clones. The expected size for positive inserts was close to 800 bp. To confirm the proper insertion and reading frame in the phagemid, a sequencing control was made on 10 independent positive clones.

### 4.4. Phage Display Library Infection

The final library was cultured and super-infected by Phage Helper M13K07 (ref. 18311019, Invitrogen, Waltham, MA, USA). After overnight culture in LB media supplemented with 2% glucose (at 37 °C, 200 rpm), the phage library was precipitated in PEG 20%/2.5 M NaCl (ref. Sigma 81260, Merck KGaA, Darmstadt, Germany) [12].

### 4.5. Phage Display

#### 4.5.1. Panning—Selection

Two successive rounds of biopanning were performed on a VHH phage library, with the protocol adapted from Russo Et Al. [16]. The first selection round was conducted using two independent wells of a Maxisorp microtiter plate (ref. 243656, Thermo Fisher Scientific, Waltham, MA, USA), coated overnight at 4 °C in PBS with 0.1 µg/well and 1 µg/well of our recombinant mIFN-γ (ref. 210127, Diaclone, Besançon, France). A negative control well without protein was included. Three additional wells were coated with Panning Block solution PBS-1% skimmed milk to pre-absorb the non-specific phages. An amount of 10 μL of the VHH phage library, pre-incubated for 30 min at RT in Panning Block solution, was added to wells containing 1 µg, 0.1 µg, or 0 µg of recombinant mIFN-γ and incubated for 60 min at RT with gentle shaking. Wells were washed eight times with PBS-Tween 0.5% buffer. Bound phages were eluted using 150 µL of Trypsin 1 mg/mL (ref. Sigma T7168-20TAB, Merck KGaA, Darmstadt, Germany). The second round of biopanning followed the same procedure, except that 3 µL of the phage pool recovered from the first round was used as input, the incubation step was performed for 30 min at RT with gentle shaking, and the washing stringency was increased to 15 washes with PBS-Tween 0.5%. After each round, TG1 cells were infected with the eluted phages, and titers were determined by spotting 5 µL of serial dilutions of the infected TG1 culture onto LB agar/Ampicillin plates [11,12]. To quantify the selection and enrichment of the eluted phages expressing anti-mIFN-γ VHH, the specific enrichment ratio was calculated after each round. This ratio represents the number of phages eluted from immobilized mIFN-γ relative to the number of phages eluted in the absence of mIFN-γ, reflecting the proportion of specific versus non-specific binders.

#### 4.5.2. Screening

Ninety-four VHH clones, randomly selected from the first and second panning rounds, were cultured in 1 mL LB medium in deep-well plates. VHH periplasmic expression was induced by adding 1 mM IPTG (ref. EU0008, Euromedex, Souffelweyersheim, France), and cultures were grown overnight at 26° [11,12]. Periplasmic extracts containing the expressed VHHs were harvested and used for screening by ELISA. To identify murine IFN-γ-specific clones and evaluate their potential as detection antibodies in a sandwich ELISA format, screening was conducted in parallel on immobilized murine IFN-γ (ref. 210127, Diaclone, Besançon, France; 50 ng) and on the complex formed by the commercial anti-rat IFN-γ capture antibody (Clone DB-1, ref. CT032, U-Cytech, Utrecht, The Netherlands; 1 µg) with murine IFN-γ (11 ng). Plates were washed twice and blocked with PBS-BSA 1–10% sucrose for 2 h at RT. Next, 20 µL of periplasmic extract containing VHHs and 80 µL of PBS-BSA 1% were added to each well and incubated for 1 h at RT. Plates were washed four times and then incubated with 100 µL of anti-Myc-Horseradish peroxidase (HRP) antibody (ref. A190-105P, Bethyl Laboratories, Montgomery, TX, USA) for 1 h at RT. After four additional washes, plates were revealed by adding 100 µL of TMB substrate (ref. K-Blue Advanced TMB Substrate 319,177 Neogen, Lansing, MI, USA). The reaction was stopped by adding 100 µL of H_2_SO_4_, and absorbance was measured at 450 nm using a BioTek ELx 808 plate reader (Agilent, Santa Clara, CA, USA).

### 4.6. Recombinant Engineering

#### 4.6.1. Candidates Identification

Based on OD measurements obtained on ELISA VHH screening, the best 48 candidates were analyzed by sequencing to determine nucleotide and amino acid sequences and eliminate redundant clones. Based on CDR3 domain analyses, clones were classified into various families.

#### 4.6.2. Reformatting and VHH Fusion hFc Recombinant Production

Selected VHH candidates were reformatted in VHH-hFc. The human IgG1 heavy constant part used for reformatting was amplified by PCR from cDNA generated from one of our existing human clones with a couple of specific primers (5′ to 3′) FC-hum-for: CCCAAATCTTGTGAC (forward) and FC-hum-rev: TTTACCCGGAGACAGGGAG (reverse). The 231 amino acid sequence coded (Pro100-Lys330) was compared to databases (https://blast.ncbi.nlm.nih.gov/Blast.cgi?PROGRAM=blastp&PAGE_TYPE=BlastSearch&LINK_LOC=blasthome (accessed on 4 December 2025)) and was found to be 100% identical to the IgG1 constant region from Homo sapiens (accession number AXN93647.1).

To produce VHH-hFc for testing purposes, the complete DNA chains (variable + constant domains) of chosen candidates were subcloned in a licensed mammalian expression vector (Figure 2). The transient production in CHO cells was outsourced at RD-Biotech company (Besançon, France). Shortly, a 50 mL suspension of CHO cells was transfected with the plasmid, and supernatants were collected 14 days post-transfection.

Transfection supernatants were purified on a 1 mL HiTrap MabSelect PrismA protein A column (ref. 17549851, Cytiva, Marlborough, MA, USA). The purified antibody concentration in mg/mL was determined on a Nanodrop 2000 (Thermo Fisher Scientific, Waltham, MA, USA) using an extinction coefficient of 1.4. The quality of purified VHH-hFC was visually assessed after electrophoresis on an SDS-PAGE gel, 4–15%.

### 4.7. Antibody Pairing Tests

#### 4.7.1. By ELISA Method

ELISA was performed using microwell plates (ref. 46866, Thermo Scientific, Waltham, MA, USA), using the commercial antibody (Clone DB-1, ref.CT032, U-Cytech, Utrecht, The Netherlands) as the capture antibody and our hFC-VHH candidates as capture antibody or detection antibody after biotinylation. The following protocol was used: Capture antibodies were coated at 1 µg/well on PBS overnight at 4 °C. Wells were blocked for 2 h at RT with PBS-BSA 1–sucrose 10%. Recombinant murine IFN-γ (ref. 210127, Diaclone, Besançon, France) or native murine IFN-γ from cell culture supernatants (see Section 4.7.3) was added at 0.5 ng/well and incubated for 2 h at RT. Plates were washed four times with 200 µL PBS-1% Tween. Biotinylated detection antibodies (100 ng/well) were added and incubated for 1 h at RT followed by four washes. Finally, 150 ng/well of Streptavidine-HRP (ref. CJ30H Agilent, Santa Clara, CA, USA) was added for 30 min at RT. Then, four washing steps were performed, and 100 µL/well of TMB (Tetramethylbenzidine) substrate (ref. K-Blue Advanced TMB Substrate 319177, Neogen, Lansing, MI, USA) was added for 15 min. Finally, the reaction was stopped with 100 µL H_2_SO_4_. OD_450nm_ was measured with the BioTek ELx808 absorbance plate reader (Agilent, Santa Clara, CA, USA).

#### 4.7.2. By ELISpot Method

The ELISpot protocol used the same antibodies as described in Section 4.7.1, but the plates, reagents, and incubation times were different. After membrane permeabilization with 25 µL/well of 35% ethanol for 30 s, Multiscreen^®^ 96 well Plate hydrophobic polyvinylidene fluoride (PVDF) membrane (ref. MSIPN, Merck KGaA, Darmstadt, Germany) was washed thoroughly five times with 250 µL/well of PBS. Then, antibodies were coated with 1 µg/well of PBS overnight at 4 °C. The next day, the wells were blocked for 2 h with PBS milk 2%. During this time, cells were prepared at the right concentration (from 100,000 to 6250 cells/wells) with the appropriate culture media. The plate was washed three times with 200 µL/well of PBS, and 100 µL of stimulated or unstimulated cells were added to the wells and incubated for 15–20 h in a CO_2_ incubator at 37 °C. After emptying the wells and removing the excess solution, 200 µL of PBS was added, and the plate was incubated at 4 °C for 10 min. Then, to ensure cell removal, the plate was washed three times with 200 µL of PBS. Then 100 µL of detection antibodies at a range of different concentrations (from 100 ng/well to 25 ng/well) were added and incubated for 1.5 h at RT. After a new washing step, 100 µL of diluted streptavidin-Alkaline phosphatase (AP) conjugate (ref. 3310-10-1000, Mabtech, Nacka Strand, Sweden) was added and incubated for 1 h at RT. Finally, the plate was washed three times with 200 µL of PBS, and the plate bottom was peeled off afterward. The membrane was thoroughly washed under running distilled water, and any excess solution was removed by repeated tapping on absorbent paper.

An amount of 100 µL of ready-to-use 5-bromo-4-chloro-3-indolyl-phosphate/nitro blue tetrazolium (BCIP/NBT) substrate (ref. NBTH-1000, Moss Inc., Pasadena, MD, USA) was added and incubated for 5–15 min away from light. After the spot apparition, a final thorough wash was performed with distilled water. After full drying, the plate was analyzed using CTL ImmunoSpot S6 Analyzer (ImmunoSpot/Cellular Technology Limited, Cleveland, OH, USA).

#### 4.7.3. Splenocyte Culture and Native Protein Production

Spleen from BALB/c mouse (Charles River, Wilmington, MA, USA) was isolated and crushed mechanically through a cell strainer. After counting, cells were cultured for 2 days in complete medium of RPMI + 5% L-Glutamine + 5% Penicillin-Streptomycin (ref. Sigma R0883, G7513, P0781, Merck KGaA, Darmstadt, Germany), at 2.10^6^ cells/mL with mIL-2 (ref. 402-ML, R&D Systems Inc., Minneapolis, MN, USA) used at 5 ng/mL and concanavalin-A (ref. Sigma C0412, Merck KGaA, Darmstadt, Germany) at 0.5 µg/mL. After the incubation, cells were collected, counted, and put in an ELISpot plate or in a flask at 1 × 10^6^ cells/mL with or without stimuli PMA (ref. Sigma P158, Merck KGaA, Darmstadt, Germany) with 5 ng/mL Ionomycin (ref. Sigma I0634, Merck KGaA, Darmstadt, Germany) at 500 ng/mL.

## Data Availability

Data is contained within the article. The original contributions presented in this study are included in the article. Further inquiries can be directed to the corresponding author.

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
