# Peer review of "Ilama VHH as a Substitute for Rabbit Polyclonal Antibodies in ELISpot Application"

_ijms, 2025, doi:10.3390/ijms262411881_

Round 1

Reviewer 1 Report

Comments and Suggestions for Authors

This manuscript talks about generating a monoclonal VHH antibody (through panning immunized library) that can serve as a detection antibody for quantifying murine IGN-gamma in an ELISpot assay.

The series of experiments leading to the identification of such an antibody seems logical. This work is scientifically sound (few modifications/clarifications are needed) and elicits interest to readers.

I however have some queries and it would be great if the authors could address them promptly.

  1. Line 126/127: Any reason authors did not perform NGS to determine library diversity? Authors mention in Table 1 that CDR3 diversity > 95%. Sample size (10 clones by Sanger sequencing) is very small to make an assumption on CDR3 diversity. Either remove that from the table or perform an NGS to comment on diversity.
  2. Table 2: Authors claim that in round 2 for PBS-BSA 1% arm, only 1 colony grew in output plate. This means that literally all the clones were specific to mIFNg after phage rescue from round 1 (R1). Is this is the case, how come authors do not see an enrichment in output numbers from mIFNg arm in R2 (in fact the value slightly decreased from R1). Can authors comment on the input titers (for both R1 and R2)?
  3. Line 183: Is it the same capture antibody that authors talked about in line 80/81? 
  4. With a specific enrichment ratio of 120000, how come only 14% of the clones were identified as binders to mIFNg? Can authors comment on the rest of the 86%? Were they non-specific binders?
  5. Line 206: Why did authors pick all 27 and not 23 (14+9) binders - that actually worked in sandwich ELISA? Any rationale?
  6. Line 212: How do authors define unique? Does change in a single AA constitute as unique? If yes, then how man y of those 24 sequences were counted unique due to somatic hypermutations in framework regions?
  7. Table 3: VHH sub-family 2 and 3 look to be very identical. It seems they belong to the same clonotype. Can authors claim them as 2 distinct families? Were the CDR1/2 were very different?
  8. Line 228: Quantity looks to be on the lower side. Were the sequences CHO-codon optimized? What kind of transfection was performed (PEI)? Can authors provide more details here
  9. Table 4: Can authors report the concentration as g/L
  10. Line 252/253: Do authors have a rationale for this observation? They seem to be binding in primary ELISA.
  11. Line 282/282: How come G4/H3 pair doesn't work in ELISA but appears to work in ELISpot pairing assay? Can authors provide explanation?
  12. Figure 5: How does the non-stimulated control look here for H3. G4 looks a cleaner option than H3. Is there a reason that authors carried H3 as well ahead?
  13. Table 6: Would be really good if authors can include HPLC data as well a QC metric. Would be interesting to see monomer content of these samples on HPLC.
  14. Can authors comment on affinities (mainly off-rate) of G4 and D-B1 binders?

Thank you.

Reviewer 2 Report

Comments and Suggestions for Authors

The authors prepared llama anti-mouse IFN-γ VHH library and screened them by ELISA. After constructing VHH-hFc, anti-mouse IFN-γ antibodies were produced in CHO cells. Although the results are very promising, I’m not sure about the novelty of the work. Specific comments follow.

Major points:

  1. Line 183: Please explain why the authors used anti-rat IFN-γ antibody instead of anti-mouse IFN-γ antibody.
  2. Line 562: Please justify to use PBS instead of Water to remove cells.

Minor points:

  1. Line 286: “Unstimulated, NS” should be “Unstimulated, US”.
  2. Line 396: “Freund” should be “Freund’s”.
  3. Line 515: Please fix “Shoy” as it doesn’t make sense.
  4. Please use consistent description for “mL” and “DB-1” throughout the manuscript.
  5. Line 609: Please arrange abbreviations alphabetical order for easier finding.

Round 2

Reviewer 2 Report

Comments and Suggestions for Authors

4. Please use consistent description for “mL” and “DB-1” throughout the manuscript.

Changes made

See lines 264, 267, 269, Table 5, 287, 292, 294, 299, 301, 306, 313, 343, 350, 351, 352,

Also, Line 34: “CD4+ and CD8+” should be “CD4+ and CD8+”.

Author Response

4. Please use consistent description for “mL” and “DB-1” throughout the manuscript.

Changes made

See lines 264, 267, 269, Table 5, 287, 292, 294, 299, 301, 306, 313, 343, 350, 351, 352,

Response: Sorry for misunderstanding concerning DB-1 notation. All replacements have been made in the manuscript on lines specify but also in Table 5, Figure 4 and Figure 5.

Also, Line 34: “CD4+ and CD8+” should be “CD4+ and CD8+”.

Response: changes have been made

The novelty of the manuscript

Response: Our article is the first one using a one single VHH antibody for the replacement of polyclonal sera in ELISpot application. And this single antibody is able to increase global performance of our ELISpot kit

Round 3

Reviewer 2 Report

Comments and Suggestions for Authors

Well done. I have no further comment.